# Dynamic Modeling and Anti-Disturbing Control of an Electromagnetic MEMS Torsional Micromirror Considering External Vibrations in Vehicular LiDAR

**DOI:** 10.3390/mi12010069

**Published:** 2021-01-09

**Authors:** Yong Hua, Shuangyuan Wang, Bingchu Li, Guozhen Bai, Pengju Zhang

**Affiliations:** 1School of Mechanical Engineering, University of Shanghai for Science and Technology, Shanghai 200093, China; huayong1997@163.com (Y.H.); bguozhen@163.com (G.B.); 2Deep Sea High-End Equipment Complex System Research Institute, University of Shanghai for Science and Technology, Shanghai 200093, China; zhangpj@lsea-tech.com

**Keywords:** electromagnetic MEMS torsional micromirror, sliding mode control, PID control, MEMS-based LiDAR, vibration suppression

## Abstract

Micromirrors based on micro-electro-mechanical systems (MEMS) technology are widely employed in different areas, such as optical switching and medical scan imaging. As the key component of MEMS LiDAR, electromagnetic MEMS torsional micromirrors have the advantages of small size, a simple structure, and low energy consumption. However, MEMS micromirrors face severe disturbances due to vehicular vibrations in realistic use situations. The paper deals with the precise motion control of MEMS micromirrors, considering external vibration. A dynamic model of MEMS micromirrors, considering the coupling between vibration and torsion, is proposed. The coefficients in the dynamic model were identified using the experimental method. A feedforward sliding mode control method (FSMC) is proposed in this paper. By establishing the dynamic coupling model of electromagnetic MEMS torsional micromirrors, the proposed FSMC is evaluated considering external vibrations, and compared with conventional proportion-integral-derivative (PID) controls in terms of robustness and accuracy. The simulation experiment results indicate that the FSMC controller has certain advantages over a PID controller. This paper revealed the coupling dynamic of MEMS micromirrors, which could be used for a dynamic analysis and a control algorithm design for MEMS micromirrors.

## 1. Introduction

With the rapid development of unmanned driving and intelligent assisted driving technology, the vehicle-mounted laser LiDAR, which serves as an important sensor for vehicle intelligent perception systems, is generating increasing interest. Vehicle-mounted LiDAR lasers have historically been split between three camps: the solid-state LiDAR, the mechanical rotating LiDAR, and MEMS-based torsional LiDAR. Solid-state LiDAR does not require rotating mechanical parts and has the advantage of being maintenance free and high resolution. However, its high price prevents solid-state LiDAR from being widely used. Tiny micromirrors are used to reflect the laser from the laser source in MEMS-based LiDAR; omni-directional scanning can be realized through the motion of the micromirror. MEMS-based LiDAR has numerous advantages, including small volume and low cost. As the main components of MEMS-based LiDAR, based on different actuation methods, MEMS micromirrors can be divided into four main types: electromagnetic [1], piezoelectric [2], electrothermal [3], and electrostatic [4]. Due to the advantages of having a large driving force, small driving voltage, and wide scanning range, the electromagnetic MEMS torsional micromirror is also an important optical device for the development of advanced MEMS-based LiDAR [5,6].

Ji [7] designed an addressable 4 × 4 array of micromirrors which was capable of providing up to 90° of angular deflection; the designed structure can be used in microphotonic applications. In 2007, Ji [8] designed a new electromagnetic scanning micromirror which is actuated by a radial magnetic field; this scanner can be used in raster scanning laser display systems. In 2009, Cho [9] designed a new three-axis electromagnetically-actuated micromirror, which has a low voltage and high coupling efficiency. Yeow [10] first designed and manufactured an electromagnetic MEMS micromirror based on hard magnetic material. Jiang [11] designed a micromirror structure with embedded coils. When the depth of the coils increased, the design has the potential to achieve larger driven forces.

Though the electromagnetic MEMS torsional micromirror can solve the problems of LiDAR miniaturization and cost, other problems still need to be solved urgently. As a vehicle-mounted instrument, the MEMS micromirror is seriously impacted by vibration originating from the vehicle body due to the rolling road surface, and the resonance of the micromirror can disturb the predefined motion trajectory of MEMS micromirror itself.

Iskiman et al. [12] presented a dynamic model of MEMS micromirrors driven by soft magnetic thin film, and in this paper, the working principle and electromagnetic force were discussed in detail. Qin et al. [13] carried out mathematical modeling and analysis on another MEMS micromirror driven by hard magnetic materials. To better control the deflection of the micromirror, Tan et al. [14] proposed a nonlinear model based-angular control strategy for electromagnetically actuated micromirrors. Cao et al. [15] proposed a data-driven modeling method for electromagnetically actuated micromirrors in a random noisy environment; the electromagnetic micromirror is considered a dynamic system with preceded hysteresis. However, existing research has focused on the static model of the MEMS micromirror, and a dynamic model of the micromirror under vibrational disturbance is still required.

To further improve the performance of the electromagnetic MEMS torsional micromirror, many scholars have studied the control method of the MEMS micromirror, which is essential for a satisfactory performance. In 2000, Pannue [16] successfully applied the proportion-integral-derivative (PID) control method to the electromagnetic MEMS micromirror, which effectively improved its dynamic response performance. Zhou et al. [17] introduced a sliding mode control based on exponential switching law, and simulation experiments proved that this strategy can effectively reduce the response time of the micromirror. At the same time, Chen et al. [18] designed an integral sliding mode control algorithm, and their experimental results proved that this strategy can achieve high precision tracking of triangle and sine waves. Compared with the PID control algorithm, the integral sliding mode control algorithm has more advantages. Considering the problem of input saturation, Tan et al. [19] designed a controller based on composite nonlinear feedback and the integral sliding mode technique, which can achieve precise positioning with a satisfactory transient performance compared with the open-loop’s performance. Based on the internal principle, a robust controller was designed in Tan’s work [20].

After investigating the literature, we found that the control research of electromagnetic MEMS micromirrors has been performed under the static state of the micromirror itself; however, due to the lack of research on dynamic modeling of the MEMS micromirror with external vibration, the performance of control methods under realistic disturbance conditions has not been evaluated.

In this paper, we proposed a dynamic coupling model of the MEMS micromirror considering external vibration, and an experimental method is used to identify the coefficients in the dynamic coupling model. The structure of this paper is as follows. In Section 1, the background of the electromagnetic MEMS torsional micromirror is introduced, including its modeling and control. Section 2 introduces the dynamic model of the electromagnetic MEMS torsional micromirror, including the working principle of the MEMS micromirror and the dynamic model considering external vibration. Section 3 introduces the anti-disturbance controller design. Conclusions are drawn in Section 4.

## 2. Dynamic Model of the Electromagnetic MEMS Torsional Micromirror

### 2.1. Working Principle of the Electromagnetic MEMS Torsional Micromirror

The structure of the electromagnetic MEMS torsional micromirror is shown in Figure 1a. It is composed of a mirror, torsion beam, permanent magnet, substrate, micromirror frame, and drive coil. The micromirror frame layer is provided with driving coils of the slow axis and the fast axis. When there is no driving signal loaded on the driving coil, the micromirror maintains a balanced state through its own gravity and the elastic force of the torsion beam. When the driving signal is loaded in different coils, the driving coil of the slow axis and the permanent magnet generate an electromagnetic force, which deflect the mirror around the slow axis. The fast axis is deflected around the fast axis by the electromagnetic force, which is generated by the permanent magnet and the driving coil on its edge. When the light beam is incident on the mirror surface, since the deflection angle of the mirror is constantly changing, the light beam can be an emitter to different positions to achieve the function of LiDAR sensing of the surrounding environmental elements. Figure 1b shows the magnetic field distribution of the electromagnetic MEMS torsional micromirror and the principle of laser scanning.

### 2.2. Dynamic Model of the Electromagnetic MEMS Torsional Micromirror

When the electromagnetic MEMS torsional micromirror is deflected by electromagnetic force, the micromirror twists around the torsion beam but does not deform itself; the deformation mainly occurs in the torsion. The dynamic deflection model can be described by the second-order vibration system of the mass-spring-damper, as follows [12]:(1)$$J_{m}\overset{\cdot \cdot }{\theta }+b\overset{\cdot }{\theta }+k_{\theta }\theta =T_{\mathrm{field}}$$
where $\overset{\cdot \cdot }{\theta }$ is the micromirror angular acceleration, $\overset{\cdot }{\theta }$ is the micromirror rotating angular velocity, θ is the micromirror rotating angle, $J_{m}$ is the moment of inertia, b is the damping factor, $k_{\theta }$ is the spring coefficient, and $T_{\mathrm{field}}$ is the total electromagnetic driving torque. $T_{\mathrm{field}}$ can be simplified as:(2)$$T_{\mathrm{field}}=\mathrm{VMH}\cos(\theta )$$
where V is the volume of the magnetic material, M is the saturation magnetization, and H is the external magnetic field strength. According to Equation (1), the electromagnetic MEMS torsional micromirror system can finally be written as:(3)$$G(s)=\frac{T_{\mathrm{field}}}{J_{m}\overset{\cdot \cdot }{\theta }+b\overset{\cdot }{\theta }+k_{\theta }\theta }$$

According to the working mode of the fast axis and the slow axis, the electromagnetic MEMS torsion micromirror can be divided into the resonant type and non-resonant type. In this manuscript, the fast axis of the electromagnetic MEMS torsional micromirror worked at the resonant frequency, and the slow axis worked at the free frequency [6]. When the micromirror is in working condition, the torsion model of the fast axis is nonlinear, and the torsion model of the slow axis is linear. Considering the practical application of electromagnetic MEMS micromirror in LiDAR, due to the high working frequency of the fast axis, only the dynamic model analysis of the slow axis is generally carried out. Its parameters are shown in Table 1. A step signal with an amplitude of 0.4 V was input into the slow axis of the micromirror through an open-loop drive, and the step response signal of the slow axis of the micromirror was collected, according to Equation (3). The system transfer function can be obtained by using the MATLAB system identification toolbox. It can be written as follows:(4)$$G(s)=\frac{-5.0416}{0.0000013626s^{2}+0.00003056s+1}$$

Figure 2 shows the comparison between the identified model and the experimental results, the change trends of the two curves are basically the same. The identified model can provide adequate insight and is sufficient for the purpose of controller design. It should be noted that some data points with large fluctuations on the experimental data curve are due to the presence of an interference signal at certain moments in the open-loop test experiment, which caused an abnormal response of the micromirror, but did not have much effect on our experiment.

### 2.3. Dynamic Model of the Electromagnetic MEMS Torsional Micromirror Considering External Vibrations

Equation (4) is the static model of the MEMS micromirror. To explore the dynamic coupling model of the MEMS micromirror considering external vibration, it is necessary to clarify whether the system model parameters will be perturbed under external vibration. Therefore, in the physical experiment, to explore the coupling response of the electromagnetic MEMS torsional micromirror and the possible change of the model parameters, external vibrations of different accelerations were loaded through an electric vibration table to simulate the vibration interference caused by an uneven road surface, which unmanned vehicles may encounter on actual road. In the research of this article, we only considered road interference with unmanned vehicles at low frequencies of 200 Hz and below.

#### 2.3.1. Overview of the Platform

The experimental framework used in the experiment is shown in Figure 3. The experimental platform designed in this paper is composed of an electromagnetic MEMS micromirror, NI CompactRIO-9024, FPGA board, voltage-controlled current amplifier circuits (VCCA), differential amplifier circuits (DAC), a laser generator, and an electric vibration table. The FPGA board is used for data communication with the upper computer, and the entire micromirror system is controlled by the NI LabVIEW program. The function of the voltage-controlled current amplification circuit module is to convert the voltage signal output from the FPGA into a current signal in the proportion required to drive the micromirror, then the micromirror torsional beam can be twisted under the driving of the Lorentz force. The driving voltage required by the electromagnetic MEMS micromirror itself is relatively small; its structure size is at the micron level and the feedback position signal is also relatively small; therefore, we added a differential amplifier circuit module to the feedback signal collection link, which effectively reduced the influence of the experimental environment on the feedback signal interference. The laser generator is used to visually observe the control effect and the response of the micromirror during the frequency sweeping operation of the electric vibration table. It is also used to simulate LiDAR, which can enlarge the effect of measurement. The electric vibration table is used to sweep the frequency of the electromagnetic MEMS torsional micromirror and provide external vibration interference. Figure 4 shows the hardware configuration of the test bench.

#### 2.3.2. Experimental Setup

When the electromagnetic MEMS torsional micromirror is used with LiDAR, to ensure a high resolution the actual driving signal of the slow axis of the micromirror is generally a 20 Hz drive signal. Therefore, the driving signal of the micromirror in this experiment was a 20 Hz sine wave signal. The sampling frequency was set to 20 kHz, and the sampling times were 1 s. In the open-loop test, the driving signal of the input slow axis was a 20 Hz sine wave signal of 0.1 V.

The acceleration of the electric vibration table was set to 0.1 g when it was used to sweep the frequency of the micromirror. For the frequency sweeping result to be closer to the real mode of the micromirror, the frequency of the electric vibration table during the frequency sweep was at a superimposed speed of 1 Hz. When the unmanned vehicle is driving on the actual road, only low-frequency vibration interference below 200 Hz is considered; therefore, the frequency was only swept to 200 Hz in the test. Since the rigid fixing structure for fixing the micromirror itself has resonance characteristics, in the actual frequency sweep and standing frequency experiments, to ensure the safety of the micromirror, the electric vibration table of acceleration was set to 0.8 g, 1.0 g, and 1.2 g, which can prevent the micromirror chip from being broken by the vibrations. When performing a standing frequency vibration experiment on the micromirror, an open-loop drive signal was applied to the micromirror at the same time through the host computer. The open-loop drive signal information is shown in Table 2. The feedback signal of the micromirror was collected, and the input and output signals of the micromirror analyzed by spectrum analysis to explore the influence of external vibrations on the micromirror.

The value of the parameter g in the acceleration of the electric vibration table was 9.18 m/s^2^. During the frequency sweep of the micromirror, the laser reflected by the micromirror can assist in viewing the resonance of the micromirror. When the laser light reflected by the micromirror is the longest, the corresponding frequency at this time is the resonance frequency of the micromirror chip.

#### 2.3.3. Hardware Test Results and Discussion

Before carrying out the physical experiment, the modal analysis of the micromirror was carried out by using COMSOL Multiphysics 5.4 software (COMSOL Inc., Burlington, MA, USA). A main vibration mode within 200 Hz was obtained, which is shown in Figure 5a. When the micromirror is disturbed by low-frequency vibrations below 200 Hz, the slow axis has a relatively large twist, and the micromirror exhibits a torsional mode vibration shape of the slow axis. To verify the correctness of the simulation, we observed the infrared laser reflected by the micromirror while sweeping the frequency of the micromirror. It can be seen from Figure 5b that the main vibration mode within 200 Hz is in accordance with the experimental results; this indicates that when the micromirror is in a state of external vibration signal interference, the micromirror will undergo coupling vibration under open-loop driving if the frequency of the external vibration is at the resonance frequency of the micromirror.

Figure 6a shows the open-loop drive response of the slow axis when the micromirror is in a static state. Figure 6b shows the open-loop drive response of the slow axis when the micromirror is disturbed by external vibrations. Here, the acceleration of the electric vibration table is 0.1 g, and the standing frequency is 136 Hz. It can be clearly seen that when the micromirror is disturbed by external vibrations, the open-loop drive response of the slow axis is greatly affected.

Figure 7 and Figure 8 show the frequency spectrum analysis of the feedback signal when the electric vibration table is under different accelerations to perform standing frequency experiments on the micromirror, and the host computer simultaneously applies an open-loop drive signal with different frequencies to the micromirror. When the frequency of the external vibration interference signal is at the main mode frequency of the micromirror, the coupled vibration of the micromirror exhibits a strict linear superposition relationship. This phenomenon indicates that when the micromirror is affected by the external vibration interference signal, and the frequency of the vibration interference signal happens to be the main mode frequency of the micromirror, the system model of the micromirror will not change.

The experimental results in Figure 7 and Figure 8 show that the coupled vibration of the micromirror at the resonant frequency of 136 Hz is a linear superposition relationship. To make this finding more credible, the feedback signal spectrum analysis was performed on the micromirror under different external vibration frequencies.

Table 3 shows the spectrum analysis of the slow axis at different driving frequencies. When the driving frequency approaches the slow axis resonance frequency of 136 Hz, the slow axis has resonance. When the driving frequency crosses the resonance frequency, the response amplitude of the slow axis begins to gradually decrease.

Table 4 shows the frequency spectrum analysis of the slow axis under different accelerations and different standing frequencies of the electric vibrating table. The micromirror was only in an external vibration state, which is not driven by an open loop signal. As can be seen from Table 4, when the acceleration of the electric vibration table increases gradually, the torsion amplitude of the slow axis also increases correspondingly. When the standing frequency is the resonant frequency of the slow axis, resonance occurs in the slow axis, and then the amplitude reaches the maximum.

Table 5, Table 6 and Table 7 show the frequency spectrum analysis of the slow axis coupled with the vibration feedback signal when the vibration interference signal is generated by the electric vibration table under different accelerations. At the same time, the micromirror also accepts the open-loop control of the drive signal. As can be seen from Table 5, Table 6 and Table 7, whether it is a low-frequency vibration lower than the resonance frequency of the micromirror, or a vibration higher than the resonance frequency of the micromirror, there are only two frequency components that need to be paid attention to in the spectrum analysis of the coupled vibration of the micromirror. This shows that when the micromirror is disturbed by external vibrations, its own system model parameters will not be perturbed, and the influence of external vibration on the drive of the micromirror is linearly superimposed.

Combined with the above spectrum analysis information, the influence of vibrations on the open-loop drive of the micromirror under different accelerations is shown in Figure 9. It can be clearly seen that when the acceleration of the electric vibration table is constant, the influence of external vibrations on the coupled vibration of the slow axis presents an obvious linear change relationship, and the change trend is consistent. Therefore, when the electromagnetic MEMS micromirror is under the interference of external vibrations at the slow axis resonance frequency and its nearby frequencies, it can be further concluded that the coupled vibration of the slow axis is only a strict linear superposition of the driving influence and the external vibration; the dynamic model parameters of the mechanical system of the micromirror itself are not perturbed.

Figure 10a is the system amplitude-frequency characteristic curve diagram of the electric vibration table transmitted to the electromagnetic micromirror under different accelerations. The amplitude-frequency characteristic curve of the transmission system under different accelerations is basically the same, which indicates that the system model of the vibrations transmitted to the electromagnetic micromirror is consistent. The general mechanical transfer system can be approximated as a second-order transfer function model. It can be seen from Figure 10b that the amplitude-frequency characteristics of the system exhibit low-pass and frequency-selective characteristics. There is a peak at 136 Hz, which is closer to a linear second-order system. To determine the specific parameters of the system model of the vibrations transmitted to the micromirror, the Bode diagram of the standard second-order system was compared with the Bode diagram of the vibration transmitted to the electromagnetic micromirror under 1.2 g acceleration, as shown in Figure 10b. When the natural oscillation angular frequency of the second-order system is set to the system resonance frequency of 136 Hz, the damping ratio is set to 0.01, and the gain is set to 0.0126; the two are well fitted, as shown in Figure 10b. Therefore, the system model of the vibrations transmitted to the electromagnetic micromirror can be expressed as Equation (5).
(5)$$G_{v}(s)=\frac{0.0126}{0.000001371s^{2}+0.00002342s+1}$$

When the electromagnetic MEMS torsional micromirror works under the interference of external vibrations, the coupling vibration presents a strict linear superposition relationship. Therefore, when the external vibration is considered, the coupling dynamic model of the electromagnetic MEMS torsional micromirror can be expressed as a linear superposition of the two. This can be represented by Equation (6):(6)$$G_{c}(s)=G(s)+G_{v}(s)$$

## 3. Anti-Disturbance Controller Design

### 3.1. FSMC Controller

Sliding mode variable structure control theory first appeared in the 1950s. As a special nonlinear control method, it has been widely used because of its strong robustness and insensitivity to external disturbances. It has been successfully applied to many control fields, especially the field of MEMS-based control; the sliding mode control (SMC) method has proven to be effective [21,22,23,24]. There are two key points in the design of a sliding mode controller: the first point is to design a sliding mode surface so that the state of the system can reach the sliding mode surface and gradually stabilize in a limited amount of time; the second point is to design a controller to ensure the state of the system can reach the sliding mode surface in a limited time. From the perspective of disturbance feedforward compensation, based on the analysis of the dynamic model of the electromagnetic MEMS torsional micromirror, we designed a sliding mode controller with feedforward compensation, and its control block diagram is shown in Figure 11.

In the sliding mode controller section, the sliding mode surface has the following form:(7)$$s(t)=ce(t)+\overset{\cdot }{e}(t)$$
where e(t) is the error signal and c is an adjustable parameter used to adjust the speed at which the tracking error of the controlled system converges to zero. To prove the stability of the controlled system, we define the Lyapunov function as:(8)$$V=\frac{1}{2}s{\left(t\right)}^{2}$$

By using the second method of Lyapunov, we differentiate V with respect to time as:(9)$$\overset{\cdot }{V}=s(t)\overset{\cdot }{s}(t)$$

When the Lyapunov condition $s(t)\overset{\cdot }{s}(t)<0$ is satisfied, then the system stability of the electromagnetic MEMS micromirror is guaranteed. According to Equation (1), the derivative of the sliding surface can be obtained as:(10)$$\overset{\cdot }{s}(t)=-\frac{1}{J_{m}}T_{\mathrm{field}}+(c-\frac{b}{J_{m}})\overset{\cdot }{e}(t)-\frac{k_{\theta }}{J_{m}}e(t)+\overset{\cdot \cdot }{\theta_{d}}(t)+\frac{b}{J_{m}}\overset{\cdot }{\theta_{d}}(t)+\frac{k_{\theta }}{J_{m}}\theta_{d}(t)$$
where $\theta_{d}(t)$ is the actual output signal of the micromirror, and the control input can be given as follows:(11)$$u_{\mathrm{SMC}}(t)=u_{\mathrm{equivalent}}+u_{\mathrm{switch}}$$
where $u_{\mathrm{equivalent}}$ and $u_{\mathrm{switch}}$ are the equivalent control and the switching control. We designed an equivalent controller as Equation (12), which is obtained from Equation (10) when $\overset{\cdot }{s}(t)=0$, and it is the necessary condition for the tracking error to remain on the sliding mode surface:(12)$$u_{\mathrm{equivalent}}=(J_{m}c-b)\overset{\cdot }{e}(t)-k_{\theta }e(t)$$

To ensure $\overset{\cdot }{V}<0$, the switching controller can be designed as:(13)$$u_{\mathrm{switch}}=\varepsilon \frac{s(t)}{\left\|s(t)\right\|}+ks(t)$$
where ε>0 and k≥0 are the control gain parameters; then, the function of $\overset{\cdot }{s}(t)$ and u(t) can be written as follows:(14)$$\overset{\cdot }{s}(t)=-\frac{1}{J_{m}}((J_{m}c-b)\overset{\cdot }{e}(t)-k_{\theta }e(t)+\varepsilon \frac{s(t)}{\left\|s(t)\right\|}+ks(t))+(c-\frac{b}{J_{m}})\overset{\cdot }{e}(t)-\frac{k_{\theta }}{J_{m}}e(t)$$
(15)$$u(t)=\left(J_{m}c-b\right)\overset{\cdot }{e}(t)-k_{\theta }e(t)+\varepsilon \frac{s(t)}{\left\|s(t)\right\|}+ks(t)$$

In the function of u(t), c, ε, and k are adjustable parameters; furthermore, we can obtain the function of $\overset{\cdot }{V}$ as follows:(16)$$\overset{\cdot }{V}=s(t)\overset{\cdot }{s}(t)=-\varepsilon \left\|s(t)\right\|-k{\left\|s(t)\right\|}^{2}\leq 0$$

It has been proven that the controlled system is stable. Although the conventional sliding mode controller can achieve a satisfactory control effect, to eliminate large external interferences, the conventional sliding mode control often requires a large switching gain. At the same time, it will inevitably increase the chattering of the system. In order to improve the control performance of the system, we added a feedforward compensation link to the designed sliding mode control. By adding feedforward compensation for dynamic interference, the gain of the switching item in the sliding mode controller and the chattering of the system are reduced. The feedforward compensation controller has the following form:(17)$$u_{\mathrm{FC}}(t)=\overset{\cdot \cdot }{\theta_{d}}(t)+\frac{b}{J_{m}}\overset{\cdot }{\theta_{d}}(t)+\frac{K_{\theta }}{J_{m}}\theta_{d}(t)$$

As can be seen from Figure 11, since the designed feedforward control link considers all the disturbance signals, the micromirror may encounter and does not form a closed loop with the system itself, there is no impact on the stability of the closed-loop system of the electromagnetic MEMS micromirror. From the perspective of feedforward control, the existence of the sliding mode control can reduce the system’s accuracy requirements for the feedforward control model. From the perspective of the sliding mode control, due to the existence of feedforward control, coarse adjustments to the interference signals the micromirror may encounter in time are possible. In this way, the combination of feedforward and feedback control can effectively reduce the burden on the control system.

### 3.2. Simulation Experiments

For validating the FSMC controller and comparing the anti-interference performance of the PID and FSMC control methods, the simulation test was performed by using MATLAB/Simulink. In the simulation experiment, we used a step disturbance signal and uniform random disturbance signal as the interference signal source. There are two types of reference track signals: the step signal and triangular wave signal. The step signal is used to compare the anti-interference ability of the controller, as the signal of the MEMS LiDAR driving the electromagnetic MEMS torsional micromirror in practical applications—the 20 Hz triangular wave signal—was mainly used to test the influence of external interference on the controller’s tracking signal ability in the simulation experiment. The step signal value given in the simulation experiment was 0.2 V.

The Disturbance signal value was 30 V and started at 0.02 s, the uniform random disturbance signal value ranged from –30 V to 30 V, and the triangular wave signal output value was—0.1 V to 0.1 V. In this paper, the controller parameter tuning method used in the simulation experiment was the empirical method. For a fair comparison, we aimed to minimize overshoot and regulate time, and adjusted the PID controller and SMC controller parameters to make the dynamic process of the two control systems similar. The two controller parameters used in the simulation experiment are as follows:(1)PID controller: $K_{p}=-100$, $K_{i}=-0.2$, $K_{d}=-0.02$.(2)SMC controller: c=5100, ε=−0.11691, k=−0.0481.

Firstly, comparative simulations were carried out for the step response of the PID and FSMC controllers in the presence of a step disturbance. Figure 12 shows the simulation results of these two controllers. Obviously, the disturbance error of the FSMC controller is smaller than that of the PID controller, which means that the tracking error fluctuation range of the FSMC controller is small. Figure 13 and Figure 14, respectively, show the effects of step disturbance and uniform random disturbance on the controller when inputting a 20 Hz triangular wave signal. Both the PID controller and the FSMC controller can achieve the signal tracking target under external disturbance; this shows that both the PID and FSMC controllers have good disturbance immunity. However, from the enlarged views of Figure 13 and Figure 14, it can be seen that the FSMC control system is less affected by external disturbance, while the PID control system is more affected by external disturbance, and the position tracking error of the PID controller is larger than that of the FSMC controller, which shows that the anti-interference ability of the FSMC controller is better than that of the PID controller.

## 4. Conclusions

In this paper, the dynamic system model of the electromagnetic MEMS torsional micromirror when there is external low-frequency vibration interference below 200 Hz is analyzed, and the coupling dynamic model of the electromagnetic MEMS torsional micromirror is established. An integrated sliding mode controller with feedforward compensation was designed, and we proposed a comparison method based on the electromagnetic MEMS torsional micromirror resonant point control. Through the experimental analysis of the performance comparison experiment of the control method and the modal analysis of the micromirror, some conclusions can be drawn. When the micromirror is disturbed by low-frequency vibrations below 200 Hz, the slow axis of the micromirror will be significantly twisted, and the coupling dynamic model of the electromagnetic MEMS torsional micromirror presents a strict linear superposition relationship at this time. The proposed FSMC control algorithm can effectively suppress the external vibrations, and has certain advantages over the PID control algorithm. The innovation of this research lies in the establishment of the coupling dynamic model of the electromagnetic MEMS micromirror under external low-frequency vibration interference below 200 Hz, and the corresponding anti-disturbance method we designed.

## Figures and Tables

**Figure 1 micromachines-12-00069-f001:**
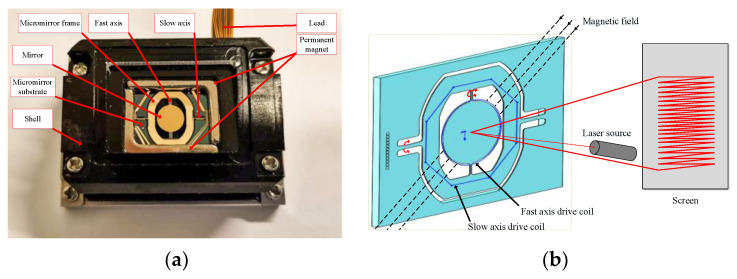
(**a**) Structure of the electromagnetic micro-electro-mechanical systems (MEMS) torsional micromirror; (**b**) principle of laser scanning.

**Figure 2 micromachines-12-00069-f002:**
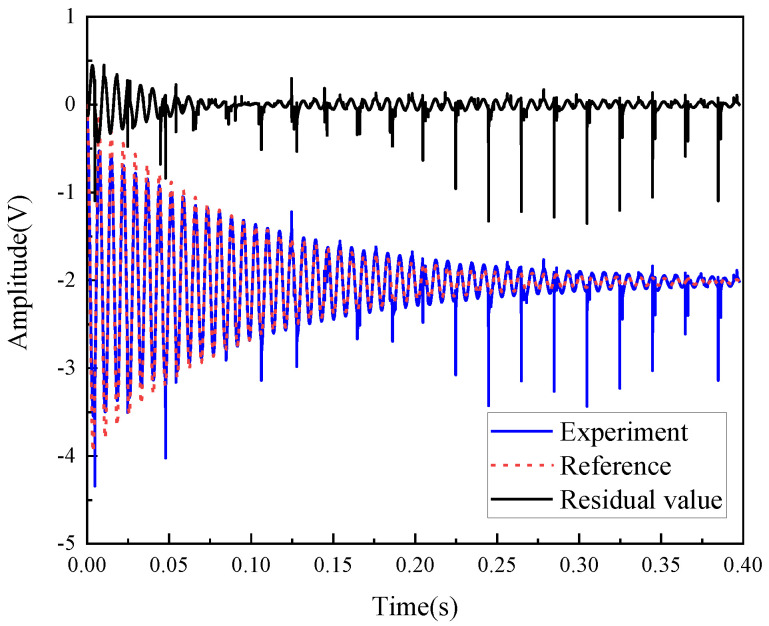
Comparison of the identification model and experiment.

**Figure 3 micromachines-12-00069-f003:**
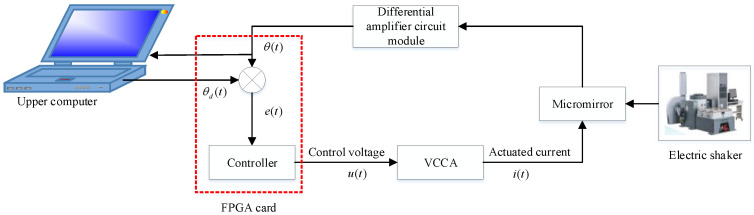
Framework of the experimental system.

**Figure 4 micromachines-12-00069-f004:**
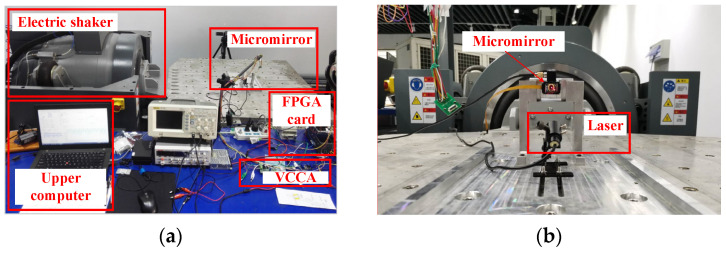
Experimental equipment diagram. (**a**) The overall layout of the experiment; (**b**) local layout of the experiment.

**Figure 5 micromachines-12-00069-f005:**
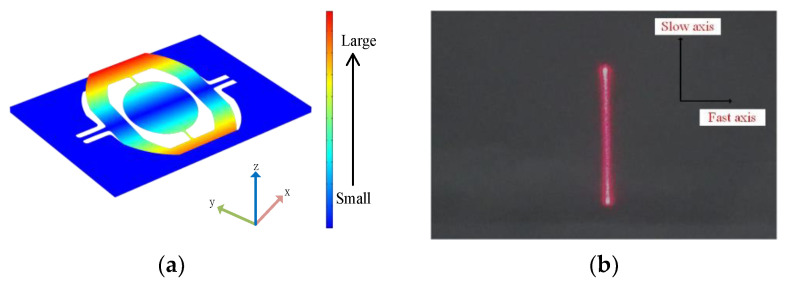
(**a**) Micromirror modal analysis diagram; (**b**) phenomenon of the micromirror sweep frequency experiment.

**Figure 6 micromachines-12-00069-f006:**
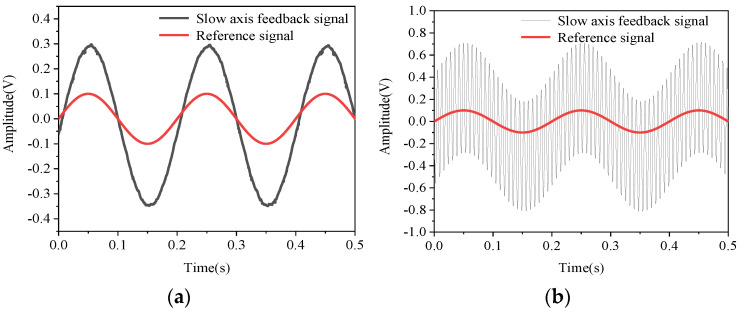
(**a**) Open-loop drive response of the micromirror under the static state; (**b**) open-loop drive response of the micromirror under vibration.

**Figure 7 micromachines-12-00069-f007:**
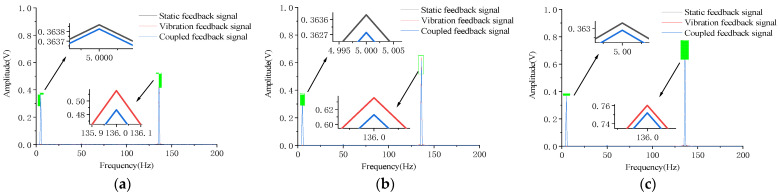
The open-loop drive signal is 5 Hz, amplitude is 0.1 V. (**a**) The acceleration of the electric vibration table is 0.8 g. (**b**) The acceleration of the electric vibration table is 1.0 g. (**c**) The acceleration of the electric vibration table is 1.2 g.

**Figure 8 micromachines-12-00069-f008:**
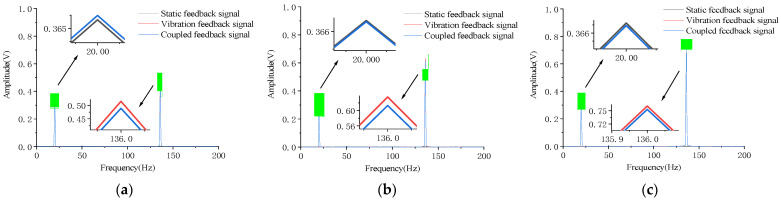
The open-loop drive signal is 20 Hz, amplitude is 0.1 V. (**a**) The acceleration of the electric vibration table is 0.8 g. (**b**) The acceleration of the electric vibration table is 1.0 g. (**c**) The acceleration of the electric vibration table is 1.2 g.

**Figure 9 micromachines-12-00069-f009:**
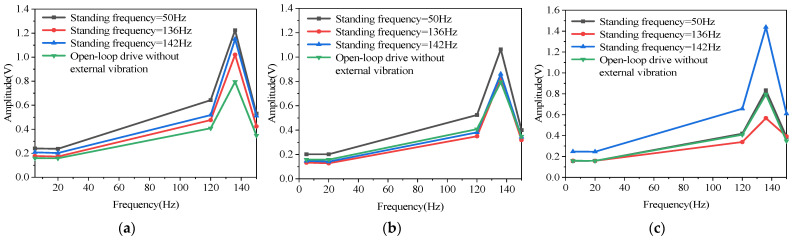
The influence of different standing frequency vibrations on the open-loop drive of the slow axis. (**a**) The acceleration of the electric vibration table is 0.8 g; (**b**) the acceleration of the electric vibration table is 1.0 g; (**c**) the acceleration of the electric vibration table is 1.2 g.

**Figure 10 micromachines-12-00069-f010:**
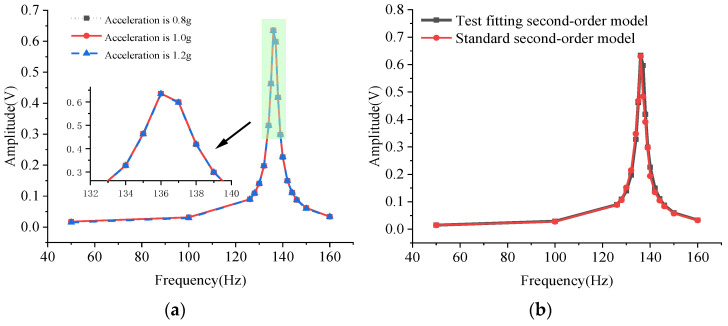
(**a**) Amplitude-frequency characteristic curve of the vibration transmission system under different accelerations; (**b**) standard second-order system Bode diagram and fitted system Bode diagram.

**Figure 11 micromachines-12-00069-f011:**
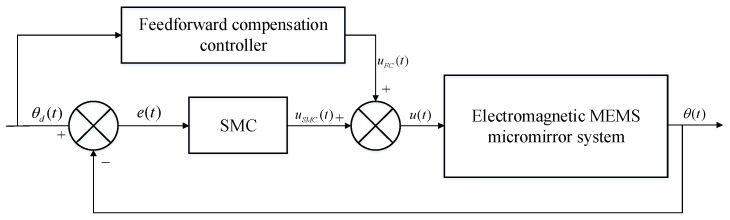
Structure of the sliding mode controller with feedforward compensation.

**Figure 12 micromachines-12-00069-f012:**
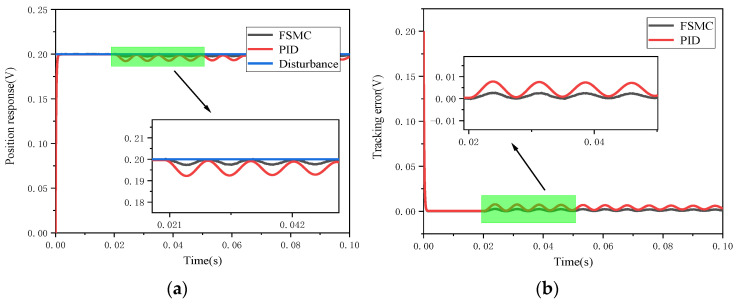
Comparison of the dynamic response performance of the electromagnetic MEMS micromirror with step signal interference. (**a**) Comparison of step tracking with step signal interference; (**b**) comparison of error convergence with step signal interference.

**Figure 13 micromachines-12-00069-f013:**
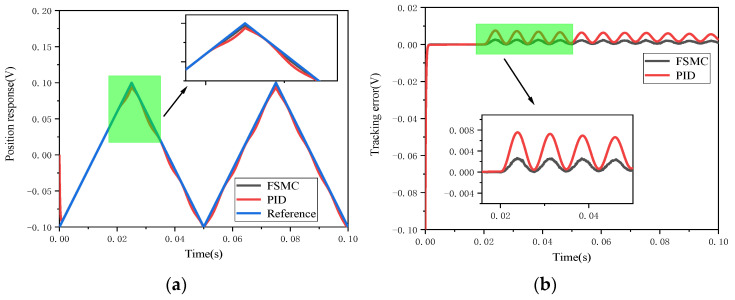
Triangular wave tracking comparison of the electromagnetic MEMS micromirror with step signal interference. (**a**) Comparison of triangular wave tracking with step signal interference; (**b**) comparison of error convergence with step signal interference.

**Figure 14 micromachines-12-00069-f014:**
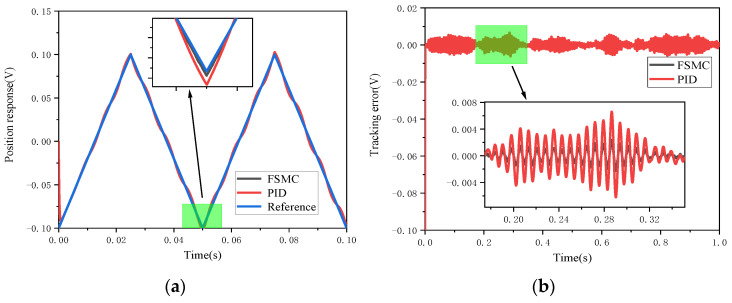
Triangular wave tracking comparison of the electromagnetic MEMS micromirror with uniform random signal interference. (**a**) Comparison of triangular wave tracking with uniform random signal interference; (**b**) comparison of triangle wave tracking error convergence with uniform random signal interference.

**Table 1 micromachines-12-00069-t001:** The parameters of the micromirror used in this paper.

Parameter	Unit	Value
Effective mirror size	mm	Φ5
Optical scanning angle	deg	30 × 20 (H × V)
Working frequency	Hz	0–170Hz (Slow axis); 1.2 K–1.3 KHz (Fast axis)
Working voltage	V	±2
Specular reflectance	%	>90
Micromirror module size	mm	25 × 16.6 × 18.2

**Table 2 micromachines-12-00069-t002:** Micromirror open-loop drive signal information.

**Frequency/Hz**	5	20	120	136	150
**Amplitude/V**	0.1	0.1	0.1	0.02	0.1

**Table 3 micromachines-12-00069-t003:** Frequency spectrum analysis of the feedback signal under the open-loop drive.

Sampling Frequency Point	Frequency (Hz)	Spectrum Data (Hz, V)
1	5	(5, 0.159)
2	20	(20, 0.158)
3	120	(120, 0.408)
4	136	(136, 0.795)
5	150	(150, 0.348)

**Table 4 micromachines-12-00069-t004:** Frequency spectrum analysis of the feedback signal under different acceleration.

Standing Frequency (Hz)	Acceleration of the Electric Vibration Table (m/s^2^)		
	0.8 g	1.0 g	1.2 g
50	(50, 0.008)	(50, 0.008)	(50, 0.010)
136	(136, 0.234)	(136, 0.234)	(136, 0.369)
142	(142, 0.048)	(142, 0.048)	(142, 0.072)

**Table 5 micromachines-12-00069-t005:** Frequency spectrum analysis of the slow axis-coupled vibration when the acceleration is 0.8 g.

Open-Loop Drive Frequency (Hz)	Standing Frequency of Electric Vibration Table (Hz)		
	50	136	142
5	(5, 0.240)	(5, 0.177)	(5, 0.208)
	(50, 0.011)	(136, 0.263)	(142, 0.063)
20	(20, 0.237)	(20, 0.173)	(20, 0.202)
	(50, 0.009)	(136, 0.258)	(142, 0.061)
120	(120, 0.642)	(120, 0.476)	(120, 0.518)
	(50, 0.010)	(136, 0.262)	(142, 0.057)
136	(136, 1.224)	(136, 1.020)	(136, 1.150)
	(50, 0.009)		(142, 0.059)
150	(150, 0.530)	(150, 0.424)	(150, 0.511)
	(50, 0.011)	(136, 0.214)	(142, 0.064)

**Table 6 micromachines-12-00069-t006:** Frequency spectrum analysis of the slow axis-coupled vibration when the acceleration is 1.0 g.

Open-Loop Drive Frequency (Hz)	Standing Frequency of Electric Vibration Table (Hz)		
	50	136	142
5	(5, 0.202)	(5, 0.133)	(5, 0.147)
	(50, 0.010)	(136, 0.250)	(142, 0.058)
20	(20, 0.202)	(20, 0.128)	(20, 0.141)
	(50, 0.010)	(16, 0.240)	(142, 0.056)
120	(120, 0.524)	(120, 0.350)	(120, 0.382)
	(50, 0.011)	(136, 0.240)	(142, 0.054)
136	(136, 1.063)	(136, 0.828)	(136, 0.861)
	(50, 0.011)		(142, 0.055)
150	(150, 0.400)	(150, 0.319)	(150, 0.344)
	(50, 0.008)	(136, 0.225)	(142, 0.055)

**Table 7 micromachines-12-00069-t007:** Frequency spectrum analysis of the slow axis-coupled vibration when the acceleration is 1.2 g.

Open-Loop Drive Frequency (Hz)	Standing Frequency of Electric Vibration Table (Hz)		
	50	136	142
5	(5, 0.157)	(5, 0.159)	(5, 0.247)
	(50, 0.008)	(136, 0.341)	(142, 0.113)
20	(20, 0.158)	(20, 0.158)	(20, 0.246)
	(50, 0.009)	(136, 0.338)	(142, 0.112)
120	(120, 0.419)	(120, 0.419)	(120, 0.657)
	(50, 0.010)	(136, 0.333)	(142, 0.110)
136	(136, 0.834)	(136, 0.567)	(136, 1.438)
	(50, 0.009)		(142, 0.110)
150	(150, 0.386)	(150, 0.392)	(150, 0.610)
	(50, 0.009)	(136, 0.309)	(142, 0.118)

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
