# Peer review of "Dynamic Modeling and Anti-Disturbing Control of an Electromagnetic MEMS Torsional Micromirror Considering External Vibrations in Vehicular LiDAR"

_micromachines, 2021, doi:10.3390/mi12010069_

Round 1
Reviewer 1 Report
- The dynamic model of the electromagnetic micromirror can be introduced with more clarity. Equation (4) is a duplicate of (3), and most equations lack details on the variables used. Line 325 has a missing variable.
- Misconception on mechanical resonance found in a few places in the article. For example: "... the fast axis .... works at the resonate frequency, and the slow axis works at the free frequency" (line 121-122). The micromirror has many resonance modes. The first two resonance modes of the mirror discussed in the paper are the rotational ones around the two pair of axis. Same as the statements in Line 234-235, and Line 243 (the slow axis has resonance).
- Editing of the English language is required; for example the paragraph between Line 229 and Line 232 is incomprehensible.
Reviewer 2 Report
This paper presents the anti-disturbing control of MEM mirror from external vibration. Authors assume that the feedback signal of the mirror represents the angle of the mirror. As written in line 150 on page 5, there is a laser generator on the experimental platform. Using the laser generator, the authors should clarify the relationship between the feedback signal and the mirror's angle.
Some minor issues
- Equation 4 is missing.
- All symbols in equations have their definition or explanation.
- In figure 2, the graph of experimental and reference are overlapped. It is hard to distinguish these two graphs.
- There are some specific values. However, there is no explanation for them.
Round 2
Reviewer 2 Report
Thank you for your kind reply.